# Differentially Private Markov Chain Monte Carlo

**Mikko A. Heikkilä** *
Helsinki Institute for Information Technology HIIT, Department of Mathematics and Statistics
University of Helsinki, Helsinki, Finland
mikko.a.heikkila@helsinki.fi

**Joonas Jälkö** *
Helsinki Institute for Information Technology HIIT, Department of Computer Science
Aalto University, Espoo, Finland
joonas.jalko@aalto.fi

**Onur Dikmen**
Center for Applied Intelligent Systems Research (CAISR)
Halmstad University, Halmstad, Sweden
onur.dikmen@hh.se

**Antti Honkela**
Helsinki Institute for Information Technology HIIT, Department of Computer Science
University of Helsinki, Helsinki, Finland
antti.honkela@helsinki.fi

## Abstract

Recent developments in differentially private (DP) machine learning and DP
Bayesian learning have enabled learning under strong privacy guarantees for the
training data subjects. In this paper, we further extend the applicability of DP
Bayesian learning by presenting the first general DP Markov chain Monte Carlo
(MCMC) algorithm whose privacy-guarantees are not subject to unrealistic assump-
tions on Markov chain convergence and that is applicable to posterior inference
in arbitrary models. Our algorithm is based on a decomposition of the Barker
acceptance test that allows evaluating the Rényi DP privacy cost of the accept-
reject choice. We further show how to improve the DP guarantee through data
subsampling and approximate acceptance tests.

## 1 Introduction

Differential privacy (DP) [Dwork et al., 2006, Dwork and Roth, 2014] and its generalisations to
concentrated DP [Dwork and Rothblum, 2016, Bun and Steinke, 2016] and Rényi DP [Mironov,
2017] have recently emerged as the dominant framework for privacy-preserving machine learning.
There are DP versions of many popular machine learning algorithms, including highly popular and
effective DP stochastic gradient descent (SGD) [Song et al., 2013] for optimisation-based learning.

There has also been a fair amount of work in DP Bayesian machine learning, with the proposed
approaches falling to three main categories: i) DP perturbation of sufficient statistics for inference in
exponential family models [e.g. Zhang et al., 2016, Foulds et al., 2016, Park et al., 2016, Bernstein
and Sheldon, 2018], ii) gradient perturbation similar to DP SGD for stochastic gradient Markov chain

---

Monte Carlo (MCMC) and variational inference [e.g. Wang et al., 2015, Jälkö et al., 2017, Li et al., 2019], and iii) DP guarantees for sampling from the exact posterior typically realised using MCMC [e.g. Dimitrakakis et al., 2014, Zhang et al., 2016, Geumlek et al., 2017].

None of these provide fully general solutions: i) sufficient statistic perturbation methods are limited to a restricted set of models, ii) stochastic gradient methods lack theoretical convergence guarantees and are limited to models with continuous variables, iii) posterior sampling methods are applicable to general models, but the privacy is conditional on exact sampling from the posterior, which is usually impossible to verify in practice.

In this paper, we present a new generic DP-MCMC method with strict, non-asymptotic privacy guarantees that hold independently of the chain's convergence. Our method is based on a recent Barker acceptance test formulation [Seita et al., 2017].

## 1.1 Our contribution

We present the first general-purpose DP MCMC method with a DP guarantee under mild assumptions on the target distribution. We mitigate the privacy loss induced by the basic method through a subsampling-based approximation. We also improve on the existing method of Seita et al. [2017] for subsampled MCMC, resulting in a significantly more accurate method for correcting the subsampling induced noise distribution.

## 2 Background

### 2.1 Differential privacy

**Definition 1** (Differential privacy). A randomized algorithm $\mathcal{M} : \mathcal{X}^N \to \mathcal{I}$ satisfies $(\epsilon, \delta)$ differential privacy, if for all adjacent datasets $\mathbf{x}, \mathbf{x}' \in \mathcal{X}^N$ and for all measurable $I \subset \mathcal{I}$ it holds that

$$\Pr(\mathcal{M}(\mathbf{x}) \in I) \leq e^\epsilon \Pr(\mathcal{M}(\mathbf{x}') \in I) + \delta. \tag{1}$$

Adjacency here means that $|\mathbf{x}| = |\mathbf{x}'|$, and $\mathbf{x}$ differs from $\mathbf{x}'$ by a single element, e.g. by a single row corresponding to one individual's data in a data matrix.

Recently Mironov [2017] proposed a Rényi divergence [Rényi, 1961] based relaxation for differential privacy called *Rényi differential privacy* (RDP).

**Definition 2** (Rényi divergence). Rényi divergence between two distributions $P$ and $Q$ defined over $\mathcal{I}$ is defined as

$$D_\alpha(P \,||\, Q) = \frac{1}{\alpha - 1} \log \mathbb{E}_P \left[ \left( \frac{p(X)}{q(X)} \right)^{\alpha - 1} \right]. \tag{2}$$

**Definition 3** (Rényi differential privacy). A randomized algorithm $\mathcal{M} : \mathcal{X}^N \to \mathcal{I}$ is $(\alpha, \epsilon)$-RDP, if for all adjacent datasets $\mathbf{x}, \mathbf{x}'$ it holds that

$$D_\alpha(\mathcal{M}(\mathbf{x}) \,||\, \mathcal{M}(\mathbf{x}')) \leq \epsilon \stackrel{\Delta}{=} \epsilon(\alpha). \tag{3}$$

Like DP, RDP has many useful properties such as invariance to post-processing. The main advantage of RDP compared to DP is the theory providing tight bounds for doing adaptive compositions, i.e., for combining the privacy losses from several possibly adaptive mechanisms accessing the same data, and subsampling [Wang et al., 2019]. RDP guarantees can always be converted to $(\epsilon, \delta)$-DP guarantees. These existing results are presented in detail in the Supplement.

### 2.2 Subsampled MCMC using Barker acceptance

The fundamental idea in standard MCMC methods [Brooks et al., 2011] is that a distribution $\pi(\theta)$ that can only be evaluated up to a normalising constant, is approximated by samples $\theta_1, \ldots, \theta_t$ drawn from a suitable Markov chain. Denoting the current parameter values by $\theta$, the next value is generated using a proposal $\theta'$ drawn from a proposal distribution $q(\theta'|\theta)$. An acceptance test is used to determine if the chain should move to the proposed value or stay at the current one.

Denoting the acceptance probability by $\alpha(\theta', \theta)$, a test that satisfies detailed balance $\pi(\theta)q(\theta'|\theta)\alpha(\theta', \theta) = \pi(\theta')q(\theta|\theta')\alpha(\theta, \theta')$ together with ergodicity of the chain are sufficient conditions to guarantee asymptotic convergence to the correct invariant distribution $\pi(\theta)$. In Bayesian inference, we are typically interested in sampling from the posterior distribution, i.e., $\pi(\theta) \propto p(\mathbf{x}|\theta)p(\theta)$. However, it is computationally infeasible to use e.g. the standard Metropolis-Hastings (M-H) test [Metropolis et al., 1953, Hastings, 1970] with large datasets, since each iteration would require evaluating $p(\mathbf{x}|\theta)$.

To solve this problem in the non-private setting, Seita et al. [2017] formulate an approximate test that only uses a fraction of the data at each iteration. In the rest of this Section we briefly rephrase their arguments most relevant for our approach without too much details. A more in-depth treatment is then presented in deriving DP MCMC in Section 3.

We start by assuming the data are exchangeable, so $p(\mathbf{x}|\theta) = \prod_{x_i \in \mathbf{x}} p(x_i|\theta)$. Let

$$\Delta(\theta', \theta) = \sum_{x_i \in \mathbf{x}} \log \frac{p(x_i|\theta')}{p(x_i|\theta)} + \log \frac{p(\theta')q(\theta|\theta')}{p(\theta)q(\theta'|\theta)}, \tag{4}$$

where we suppress the parameters for brevity in the following, and let $V_{log} \sim \text{Logistic}(0, 1)$. Instead of using the standard M-H acceptance probability $\min\{\exp(\Delta), 1\}$, Seita et al. [2017] use a form of Barker acceptance test [Barker, 1965] to show that testing if

$$\Delta + V_{log} > 0 \tag{5}$$

also satisfies detailed balance. To ease the computational burden, we now want to use only a random subset $S \subset \mathbf{x}$ of size $b$ instead of full data of size $N$ to evaluate acceptance. Let

$$\Delta^*(\theta', \theta) = \frac{N}{b} \sum_{x_i \in S} \log \frac{p(x_i|\theta')}{p(x_i|\theta)} + \log \frac{p(\theta')q(\theta|\theta')}{p(\theta)q(\theta'|\theta)}. \tag{6}$$

Omitting the parameters again, $\Delta^*$ is now an unbiased estimator for $\Delta$, and assuming $x_i$ are iid samples from the data distribution, $\Delta^*$ has approximately normal distribution by the Central Limit Theorem (CLT).

In order to have a test that approximates the exact full data test (5), we decompose the logistic noise as $V_{log} \simeq V_{norm} + V_{cor}$, where $V_{norm}$ has a normal distribution and $V_{cor}$ is a suitable correction. Relying on the CLT and on this decomposition we write $\Delta^* + V_{cor} \simeq \Delta + V_{norm} + V_{cor} \simeq \Delta + V_{log}$, so given the correction we can approximate the full data exact test using a minibatch.

## 2.3 Tempering

When the sample size $N$ is very large, one general problem in Bayesian inference is that the posterior includes more and more details. This often leads to models that are much harder to interpret while only marginally more accurate than simpler models (see e.g. Miller and Dunson 2019). One way of addressing this issue is to scale the log-likelihood ratios in (4) and (6), so instead of $\log p(x_i|\theta)$ we would have $\tau \log p(x_i|\theta)$ with some $\tau$. The effect of scaling with $0 < \tau < 1$ is then to spread the posterior mass more evenly. We will refer to this scaling as tempering.

As an interesting theoretical justification for tempering, Miller and Dunson [2019] show a relation between tempered likelihoods and modelling error. The main idea is to take the error between the theoretical pure data and the actual observable data into account in the modelling. Denote the observed data with lowercase and errorless random variables with uppercase letters, and let $R \sim \text{Exp}(\beta)$. Then using empirical KL divergence as our modelling error estimator $d_N$, instead of the standard posterior we are looking for the posterior conditional on the observed data being close to the pure data, i.e., we want $p(\theta|d_N(x_{1:N}, X_{1:N}) < R)$, which is called coarsened posterior or *c-posterior*.

Miller and Dunson [2019] show that with these assumptions

$$p(\theta|d_N(x_{1:N}, X_{1:N}) < R) \stackrel{\propto}{\sim} p(\mathbf{x}|\theta)^{\xi_N}p(\theta), \tag{7}$$

where $\stackrel{\propto}{\sim}$ means approximately proportional to, and $\xi_N = 1/(1+N/\beta)$, i.e., a posterior with tempered likelihoods can be interpreted as an approximate c-posterior.

# 3 Privacy-preserving MCMC

Our aim is to sample from the posterior distribution of the model parameters while ensuring differential privacy. We start in Section 3.2 by formulating DP MCMC based on the exact full data Barker acceptance presented in Section 2.2. To improve on this basic algorithm, we then introduce subsampling in Section 3.3. The resulting DP subsampled MCMC algorithm has significantly better privacy guarantees as well as computational requirements than the full data version.

## 3.1 Notation

There are multiple different factors that we use in the privacy analysis. Table 1 includes all the necessary factors used.

| Notation | Explanation |
|---|---|
| $\alpha$ | Parameter for RDP |
| $T$ | Number of MCMC draws |
| $N$ | Dataset size |
| $C \in (0, \pi^2/3)$ | Noise variance, in Section 3.3 we set $C = 2$ |
| $B$ | Assumed bound for the log-likelihood ratios (llr) w.r.t. data OR the parameters |
| $b > 5\alpha$ | Batch size for subsampled DP-MCMC |
| $\beta$ | Parameter for tempering |

Table 1: Table of the notation used in Section 3.

## 3.2 DP MCMC

To achieve privacy-preserving MCMC, we repurpose the decomposition idea mentioned in Section 2.2 with subsampling, i.e., we decompose $V_{log}$ in the exact test (5) into normal and correction variables. Noting that $V_{log}$ has variance $\pi^2/3$, fix $0 < C < \pi^2/3$ a constant and write

$$V_{log} \simeq \mathcal{N}(0, C) + V_{cor}^{(C)}, \tag{8}$$

where $V_{cor}^{(C)}$ is the correction with variance $\pi^2/3 - C$. Now testing if

$$\mathcal{N}(\Delta, C) + V_{cor}^{(C)} > 0 \tag{9}$$

is approximately equivalent to (5).

Since (8) holds exactly for no known distribution $V_{cor}^{(C)}$ with an analytical expression, Seita et al. [2017] construct an approximation by discretising the convolution implicit in (8), and turning the problem into a ridge regression problem which can be solved easily. Unlike Seita et al. [2017], we aim for preserving privacy. We therefore want to work with relatively large values of $C$ for which the ridge regression based solution does not give a good approximation. Instead, we propose to use a Gaussian mixture model approximation, which gives good empirical performance for larger $C$ as well. The details of the approximation with related discussion can be found in the Supplement.

In practice, if $V_{cor}^{(C)}$ is an approximation, the stationary distribution of the chain might not be the exact posterior. However, when the approximation (8) is good, the accept-reject decisions are rarely affected and we can expect to stay close to the true posterior. Clearly, in the limit of decreasing $C$ we recover the exact test (5). We return to this topic in Section 3.3.

Considering privacy, on each MCMC iteration we access the data only through the log-likelihood ratio $\Delta$ in the test (9). To achieve RDP, we therefore need a bound for the Rényi divergence between two Gaussians $\mathcal{N}_{\mathbf{x}} = \mathcal{N}(\Delta_{\mathbf{x}}, C)$ and $\mathcal{N}_{\mathbf{x}'} = \mathcal{N}(\Delta_{\mathbf{x}'}, C)$ corresponding to neighbouring datasets $\mathbf{x}, \mathbf{x}'$. The following Lemma states the Rényi divergence between two Gaussians:

**Lemma 1.** *Rényi divergence between two normals $\mathcal{N}_1$ and $\mathcal{N}_2$ with parameters $\mu_1, \sigma_1$ and $\mu_2, \sigma_2$ respectively is*

$$D_\alpha(\mathcal{N}_1 \,\|\, \mathcal{N}_2) = \ln \frac{\sigma_2}{\sigma_1} + \frac{1}{2(\alpha - 1)} \ln \frac{\sigma_2^2}{\sigma_\alpha^2} + \frac{\alpha}{2} \frac{(\mu_1 - \mu_2)^2}{\sigma_\alpha^2}, \tag{10}$$

*where $\sigma_\alpha^2 = \alpha \sigma_2^2 + (1 - \alpha)\sigma_1^2$.*

*Proof.* See [Gil et al., 2013] Table 2. □

**Theorem 1.** *Assume either*

$$|\log p(x_i \,|\, \theta') - \log p(x_i \,|\, \theta)| \leq B \tag{11}$$

*or*

$$|\log p(x_i \,|\, \theta) - \log p(x_j \,|\, \theta)| \leq B, \tag{12}$$

*for all $x_i, x_j$ and for all $\theta, \theta'$. Releasing a result of the accept/reject decision from the test* (9) *is* $(\alpha, \epsilon)$*-RDP with $\epsilon = 2\alpha B^2/C$.*

*Proof.* Follows from Lemma 1. See Supplement for further details. □

Using the composition property of RDP (see Supplement), it is straightforward to get the following Corollary for the whole chain:

**Corollary 1.** *Releasing an MCMC chain of $T$ iterations, where at each iteration the accept-reject decision is done using the test* (9)*, satisfies $(\alpha, \epsilon')$-RDP with $\epsilon' = T2\alpha B^2/C$.*

We can satisfy the condition (11) with sufficiently smooth likelihoods and a proposal distribution with a bounded domain:

**Lemma 2.** *Assuming the model log-likelihoods are $L$-Lipschitz over $\theta$ and the diameter of the proposal distribution domain is bounded by $d_\theta$, LHS of* (11) *is bounded by $Ld_\theta$.*

*Proof.*

$$|\log p(x_i \,|\, \theta) - \log p(x_i \,|\, \theta')| \leq L|\theta - \theta'| \leq Ld_\theta. \tag{13}$$

□

Clearly, when $Ld_\theta \leq B$ we satisfy the condition in Equation (11).

For some models, using a proposal distribution with a bounded domain could affect the ergodicity of the chain. Considering models that are not Lipschitz or using an unbounded proposal distribution, we can also satisfy the boundedness condition (11) by clipping the log-likelihood ratios to a suitable interval.

### 3.3 DP subsampled MCMC

In Section 3.2 we showed that we can release samples from the MCMC algorithm under privacy guarantees. However, as already discussed, evaluating the log-likelihood ratios might require too much computation with large datasets. Using the full dataset in the DP MCMC setting might also be infeasible for privacy reasons: the noise variance $C$ in Theorem 1 is upper-bounded by the variance of the logistic random variable, and thus working under a strict privacy budget we might be able to run the chain for only a few iterations before $\epsilon'$ in Corollary 1 exceeds our budget. Using only a subsample $S$ of the data at each MCMC iteration allows us to reduce not only the computational cost but also the privacy cost through privacy amplification [Wang et al., 2019].

As stated in Section 2.2, for the subsampled variant according to the CLT we have

$$\Delta^* = \Delta + \tilde{V}_{norm}, \tag{14}$$

where $\tilde{V}_{norm}$ is approximately normal with some variance $\sigma^2_{\Delta^*}$. Assuming

$$\sigma^2_{\Delta^*} < C < \pi^2/3 \tag{15}$$

for some constant $C$, we now reformulate the decomposition (8) as

$$V_{log} \simeq \underbrace{V_{norm} + V_{nc}}_{\sim \mathcal{N}(0,C)} + V_{cor}^{(C)}, \tag{16}$$

where $V_{norm} \sim \mathcal{N}(0, \sigma^2_{\Delta^*})$ and $V_{nc} \sim \mathcal{N}(0, C - \sigma^2_{\Delta^*})$. We can now write

$$\Delta^* + V_{nc} + V_{cor}^{(C)} \simeq \Delta + V_{norm} + V_{nc} + V_{cor}^{(C)} \simeq \Delta + V_{log}, \tag{17}$$

where the first approximation is justified by the CLT, and the second by the decomposition (16). Therefore, testing if

$$\mathcal{N}(\Delta^*, C - \sigma^2_{\Delta^*}) + V^{(C)}_{cor} > 0 \tag{18}$$

approximates the exact full data test (5).

As in Section 3.2, the approximations used for arriving at the test (18) imply that the stationary distribution of the chain need not be the exact posterior. However, we can expect to stay close to the true posterior when the approximations are good, since the result only changes if the binary accept-reject decision is affected. This is exemplified by the testing in Section 4 (see also Seita et al. 2017). The quality of the first approximation in (17) depends on the batch size $b$, which should not be too small. As for the second error source, as already noted in Section 3.2 we markedly improve on this with the GMM based approximation, and the resulting error is typically very small (see Supplement). In some cases there are known theoretical upper bounds for the total error w.r.t. the true posterior. These bounds are of limited practical value since they rely on assumptions that can be hard to meet in general, and we therefore defer them to the Supplement.

For privacy, similarly as in Section 3.2, in (18) we need to access the data only for calculating $\Delta^* + V_{nc}$. Thus, it suffices to privately release a sample from $\mathcal{N}_S = \mathcal{N}(\Delta^*_{\mathbf{x}}, C - s^2_{\Delta^*_{\mathbf{x}}})$, where $s^2_{\Delta^*_{\mathbf{x}}}$ denotes the sample variance when sampling from dataset $\mathbf{x}$, i.e., we need to bound the Rényi divergence between $\mathcal{N}_S$ and $\mathcal{N}_{S'}$. We use noise variance $C = 2$ in the following analysis.

Next, we will state our main theorem giving an explicit bound that can be used for calculating the privacy loss for a single MCMC iteration with subsampling:

**Theorem 2.** *Assuming*

$$|\log p(x_i|\theta') - \log p(x_i|\theta)| \leq \frac{\sqrt{b}}{N}, \tag{19}$$

$$\alpha < \frac{b}{5}, \tag{20}$$

*where $b$ is the size of the minibatch $S$ and $N$ is the dataset size, releasing a sample from $\mathcal{N}_S$ satisfies $(\alpha, \epsilon)$-RDP with*

$$\epsilon = \frac{5}{2b} + \frac{1}{2(\alpha-1)} \ln \frac{2b}{b-5\alpha} + \frac{2\alpha}{b-5\alpha}. \tag{21}$$

*Proof.* The idea of the proof is straightforward: we need to find an upper bound for each of the terms in Lemma 1, which can be done using standard techniques. Note that for $C = 2$, (19) implies that the variance assumption (15) holds. See Supplement for the full derivation. □

Using the composition [Mironov, 2017] and subsampling amplification [Wang et al., 2019] properties of Rényi DP (see Supplement), we immediately get the following:

**Corollary 2.** *Releasing a chain of $T$ subsampled MCMC iterations with sampling ratio $q$, each satisfying $(\alpha, \epsilon(\alpha))$-RDP with $\epsilon(\alpha)$ from Theorem 2, is $(\alpha, T\epsilon')$-RDP with*

$$\epsilon' = \frac{1}{\alpha-1} \log \left( 1 + q^2 \binom{\alpha}{2} \min\{4(e^{\epsilon(2)} - 1), 2e^{\epsilon(2)}\} + 2 \sum_{j=3}^{\alpha} q^j \binom{\alpha}{j} e^{(j-1)\epsilon(j)} \right). \tag{22}$$

Figures 1(a) and 1(b) illustrate how changing the parameters $q$ and $T$ in Corollary 2 will affect the privacy budget of DP MCMC.

Similarly as in the full data case in Section 3.2, we can satisfy the condition (19) with sufficiently smooth likelihoods or by clipping. Figure 1(c) shows how frequently we need to clip the log-likelihood ratios to maintain the bound in (19) as a function of proposal variance using a Gaussian mixture model problem defined in Section 4. Using smaller proposal variance will result in smaller changes in the log-likelihoods between the previous and the proposed parameter values, which entails fewer clipped values.

However, the bound in (19) gets tighter with increasing $N$. To counterbalance this, either the proposals need to be closer to the current value (assuming suitably smooth log-likelihood), resulting

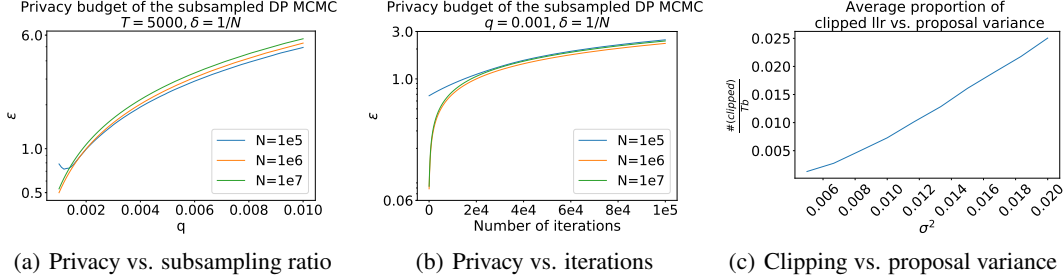

(a) Privacy vs. subsampling ratio    (b) Privacy vs. iterations    (c) Clipping vs. proposal variance

Figure 1: Parameter effects. Calculating total privacy budget from Corollary 2 for different dataset sizes: in Figure 1(a) as a function of subsampling ratio, and in Figure 1(b) as a function of number of iterations. Figure 1(c) shows the proportion of clipped log-likelihood ratios as a function of proposal variance for the GMM example detailed in Section 4.

in a slower mixing chain, or $b$ needs to increase, affecting privacy amplification. For very large $N$ we would therefore like to temper the log-likelihood ratios in a way that we could use sufficiently small batches to benefit from privacy amplification, while still preserving sufficient amount of information from the likelihoods and reasonable mixing properties. Using the c-posterior discussed in Section 2.3 with parameter $\beta$ s.t. $N_0 = N\beta/(\beta + N)$, instead of condition (19) we then require

$$|\log p(x_i|\theta') - \log p(x_i|\theta)| \leq \frac{\sqrt{b}}{N_0}, \tag{23}$$

which does not depend on $N$.

## 4   Experiments

In order to demonstrate our proposed method in practice, we use a simple 2-dimensional Gaussian mixture model[2], that has been used by Welling and Teh [2011] and Seita et al. [2017] in the non-private setting:

$$\theta_j \sim \mathcal{N}(0, \sigma_j^2,), \quad j = 1, 2 \tag{24}$$
$$x_i \sim 0.5 \cdot \mathcal{N}(\theta_1, \sigma_x^2) + 0.5 \cdot \mathcal{N}(\theta_1 + \theta_2, \sigma_x^2), \tag{25}$$

where $\sigma_1^2 = 10, \sigma_2^2 = 1, \sigma_x^2 = 2$. For the observed data, we use fixed parameter values $\theta = (0, 1)$. Following Seita et al. [2017], we generate $10^6$ samples from the model to use as training data. We use $b = 1000$ for the minibatches, and adjust the temperature of the chain s.t. $N_0 = 100$ in (23). This corresponds to the temperature used by Seita et al. [2017] in their non-private test.

If we have absolutely no idea of a good initial range for the parameter values, especially in higher dimensions the chain might waste the privacy budget in moving towards areas with higher posterior probability. In such cases we might want to initialise the chain in at least somewhat reasonable location, which will cost additional privacy. To simulate this effect, we use the differentially private variational inference (DPVI) introduced by Jälkö et al. [2017] with a small privacy budget $(0.22, 10^{-6})$ to find a rough estimate for the initial location.

As shown in Figure 2, the samples from the tempered chain with DP are nearly indistinguishable from the samples drawn from the non-private tempered chain. We also compared our method against DP stochastic gradient Langevin dynamics (DP SGLD) method of Li et al. [2019]. Figure 3 illustrates how the accuracy is affected by privacy. Posterior means and variances are computed from the first $t$ iterations of the private chain alongside the privacy cost $\epsilon$, which increases with $t$. The baseline is given by a non-private chain after 40000 iterations. The plots show the mean and the standard error of the mean over 20 runs of 20 000 iterations with DP MCMC and 6 000 000 with DP SGLD. The DP MCMC method was burned in for 1 000 iterations and DP SGLD for 100 000 iterations.

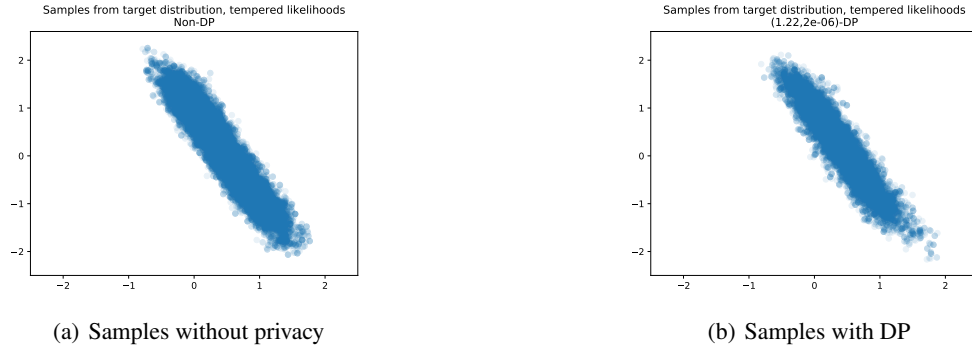

(a) Samples without privacy

(b) Samples with DP

Figure 2: Results for the GMM experiment with tempered likelihoods: 2(a) shows 40000 samples from the chain without privacy and 2(b) 20000 samples with privacy. The results with strict privacy are very close to the non-private results.

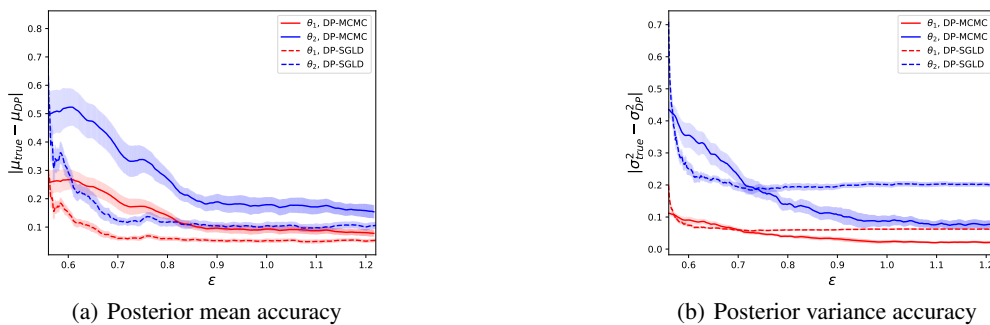

(a) Posterior mean accuracy

(b) Posterior variance accuracy

Figure 3: Intermediate private posterior statistics from DP SGLD and DP MCMC compared against the baseline given by a non-private chain after $40000$ iterations. Lines showing the mean error between 20 runs of the algorithm with errorbars illustrating the standard error of the mean between the runs. DP SGDL converges quickly towards the posterior mean, but does not properly capture posterior variance.

# 5    Related work

Bayesian posterior sampling under DP has been studied using several different approaches. Recently Yıldırım and Ermiş [2019] proposed a method for drawing samples from exact posterior under DP using a modified MH algorithm. However their solution does not include subsampling and thus suffers the computational cost of the full likelihood. Dimitrakakis et al. [2014] note that drawing a single sample from the posterior distribution of a model where the log-likelihood is Lipschitz or bounded yields a DP guarantee. The bound on $\epsilon$ can be strengthened by tempering the posterior by raising the likelihood to a power $\tau \in (0, 1)$ to obtain the tempered posterior

$$\pi_\tau(\theta) \propto p(\theta)p(\mathbf{x} \mid \theta)^\tau. \tag{26}$$

The same principle is discussed and extended by Wang et al. [2015], Zhang et al. [2016] and Dimitrakakis et al. [2017] in the classical DP setting and by Geumlek et al. [2017] in the RDP setting. Wang et al. [2015] dub this the "one posterior sample" (OPS) mechanism. The main limitation of all these methods is that the privacy guarantee is conditional on sampling from the exact posterior, which is in most realistic cases impossible to verify.

The other most widely used approach for DP Bayesian inference is perturbation of sufficient statistics of an exponential family model using the Laplace mechanism. This straightforward application of the Laplace mechanism was mentioned at least by Dwork and Smith [2009] and has been widely applied since by several authors [e.g. Zhang et al., 2016, Foulds et al., 2016, Park et al., 2016, Honkela et al., 2018, Bernstein and Sheldon, 2018]. In particular, Foulds et al. [2016] show that the sufficient statistics perturbation is more efficient than OPS for models where both are applicable. Furthermore, these methods can provide an unconditional privacy guarantee. Many of the early methods ignore the Laplace noise injected for DP in the inference, leading to potentially biased inference results. This weakness is addressed by Bernstein and Sheldon [2018], who include the uncertainty arising from

the injected noise in the modelling, which improves especially the accuracy of posterior variances for models where this can be done.

MCMC methods that use gradient information such as Hamiltonian Monte Carlo (HMC) and various stochastic gradient MCMC methods have become popular recently. DP variants of these were first proposed by Wang et al. [2015] and later refined by Li et al. [2019] to make use of the moments accountant [Abadi et al., 2016]. The form of the privacy guarantee for these methods is similar to that of our method: there is an unconditional guarantee for models with a differentiable Lipschitz log-likelihood that weakens as more iterations are taken. Because of the use of the gradients, these methods are limited to differentiable models and cannot be applied to e.g. models with discrete variables.

Before Seita et al. [2017], the problem of MCMC without using the full data has been considered by many authors (see Bardenet et al. 2017 for a recent literature survey). The methods most closely related to ours are the ones by Korattikara et al. [2014] and Bardenet et al. [2014]. From our perspective, the main problem with these approaches is the adaptive batch size: the algorithms may regularly need to use all observations on a single iteration [Seita et al., 2017], which clashes with privacy amplification. Bardenet et al. [2017] have more recently proposed an improved version of their previous technique alleviating the problem, but the batch sizes can still be large for privacy amplification.

## 6   Discussion

While gradient-based samplers such as HMC are clearly dominant in the non-DP case, it is unclear how useful they will be under DP. Straightforward stochastic gradient methods such as stochastic gradient Langevin dynamics (SGLD) can be fast in initial convergence to a high posterior density region, but it is not clear if they can explore that region more efficiently. We can see this in Figure 3: the gradient adjusted method rapidly converges close to posterior mean, but the posterior variance is not captured. HMC does have a clear advantage at exploration, but Betancourt [2015] demonstrates that HMC is very sensitive to having accurate gradients and therefore a naive DP HMC is unlikely to perform well. We believe that using a gradient-based method such as DP variational inference [Jälkö et al., 2017] as an initiasation for the proposed method can yield overall a very efficient sampler that can take advantage of the gradients in the initial convergence and of MCMC in obtaining accurate posterior variances. Further work in benchmarking different approaches over a number of models is needed, but it is beyond the scope of this work.

The proposed method allows for structurally new kind of assumptions to guarantee privacy through forcing bounds on the proposal instead of or in addition to the likelihood. This opens the door for a lot of optimisation in the design of the proposal. It is not obvious how the proposal should be selected in order to maximise the amount of useful information obtained about the posterior under the given privacy budget, when one has to balance between sampler acceptance rate and autocorrelation as well as privacy. We leave this interesting question for future work.

### Acknowledgements

The authors would like to thank Daniel Seita for sharing the original code for their paper.

This work has been supported by the Academy of Finland [Finnish Center for Artificial Intelligence FCAI and grants 294238, 303815, 313124].

## Footnotes

[2]The code for running all the experiments is avalaible in `https://github.com/DPBayes/DP-MCMC-NeurIPS2019`.

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
