[Supplementary Material · DP-MCMC-Supplement.pdf]

# Supplement to Differentially Private Markov Chain Monte Carlo

May 23, 2019

## 1 Useful differential privacy results

**Proposition 1.** A composition of two RDP algorithms $\mathcal{M}_1$, $\mathcal{M}_2$ with RDP guarantees $(\alpha, \epsilon_1)$ and $(\alpha, \epsilon_2)$, is $(\alpha, \epsilon_1 + \epsilon_2)$-RDP.

*Proof.* See Mironov [2017, Proposition 1] . □

The next result follows immediately from Proposition 1.

**Corollary 1.** *Releasing a result from a $T$-fold composition of a $(\alpha, \epsilon)$-RDP query is $(\alpha, T\epsilon)$-RDP.*

The following Proposition states the privacy amplification via subsampling result of Wang et al. [2019].

**Proposition 2.** A randomised algorithm $\mathcal{M}$ which accesses the whole dataset $\mathbf{x}$ only through subset $S$ of the dataset and satisfies $(\alpha, \epsilon)$-RDP w.r.t. to $S$, is $(\alpha, \epsilon')$-RDP with

$$\epsilon' \leq \frac{1}{\alpha - 1} \log \left( 1 + q^2 \binom{\alpha}{2} \cdot \min \left\{ 4(e^{\epsilon(2)} - 1), e^{\epsilon(2)} \min \left\{ 2, (e^{\epsilon(\infty)-1})^2 \right\} \right\} \right.$$
$$\left. + \sum_{j=3}^{\alpha} q^j \binom{\alpha}{j} e^{(j-1)\epsilon(j)} \min \left\{ 2, (e^{\epsilon(\infty)} - 1)^j \right\} \right),$$

where $q = |S|/|\mathbf{x}|$, and $\alpha \geq 2$ is an integer, and $\epsilon(\infty) = \lim_{j \to \infty} \epsilon(j)$.

*Proof.* See Wang et al. [2019, Theorem 10] . □

Finally, we can convert RDP privacy guarantees back to $(\epsilon, \delta)$-DP guarantees using the following proposition.

**Proposition 3.** An $(\alpha, \epsilon)$-RDP algorithm $\mathcal{M}$ also satisfies $(\epsilon', \delta)$-DP for all $0 < \delta < 1$ with

$$\epsilon' = \epsilon + \frac{\log(1/\delta)}{\alpha - 1}. \tag{1}$$

*Proof.* See Mironov [2017, Proposition 3] . □

# 2 Proof of main text's Theorem 1

Denote the maximally different adjacent datasets by $\mathbf{x}_1, \mathbf{x}_2$. The mechanism releases a sample from $\mathcal{N}_1 = \mathcal{N}(\Delta_1, C)$, and $\mathcal{N}_2 = \mathcal{N}(\Delta_2, C)$, where $\Delta_1, \Delta_2$ are calculated with $\mathbf{x}_1, \mathbf{x}_2$, respectively.

We want to show that

$$D_\alpha(\mathcal{N}_1 || \mathcal{N}_2) = \log \frac{\sigma_1}{\sigma_2} + \frac{1}{2(\alpha - 1)} \log \frac{\sigma_2^2}{\alpha \sigma_2^2 + (1 - \alpha)\sigma_1^2} + \frac{\alpha}{2} \frac{(\mu_1 - \mu_2)^2}{\alpha \sigma_2^2 + (1 - \alpha)\sigma_1^2} \tag{2}$$

$$\leq \frac{2\alpha B^2}{C} \tag{3}$$

assuming that either

$$|\log p(x_i | \theta') - \log p(x_i | \theta)| < B \; \forall x_i, \theta, \theta' \tag{4}$$

or

$$|\log p(x_i | \theta) - \log p(x_j | \theta)| < B, \; \forall x_i, x_j, \theta. \tag{5}$$

*Proof.* W.l.o.g., we can assume that the differing element between $\mathbf{x}_1$ and $\mathbf{x}_2$ is the final one, so $x_{1,i} = x_{2,i}, i = 1, \ldots, N - 1$.

Since $\sigma_1^2 = \sigma_2^2 = C$, we immediately have

$$D_\alpha(\mathcal{N}_1 || \mathcal{N}_2) = \log \frac{\sigma_1}{\sigma_2} + \frac{1}{2(\alpha - 1)} \log \frac{\sigma_2^2}{\alpha \sigma_2^2 + (1 - \alpha)\sigma_1^2} + \frac{\alpha}{2} \frac{(\mu_1 - \mu_2)^2}{\alpha \sigma_2^2 + (1 - \alpha)\sigma_1^2} \tag{6}$$

$$= \frac{\alpha}{2C}(\mu_1 - \mu_2)^2 \tag{7}$$

$$= \frac{\alpha}{2C} [\sum_{i=1}^N \log \frac{p(x_{1,i}|\theta')}{p(x_{1,i}|\theta)} - \sum_{i=1}^N \log \frac{p(x_{2,i}|\theta')}{p(x_{2,i}|\theta)}]^2 \tag{8}$$

$$= \frac{\alpha}{2C} \left| \log \frac{p(x_{1,N}|\theta')}{p(x_{1,N}|\theta)} - \log \frac{p(x_{2,N}|\theta')}{p(x_{2,N}|\theta)} \right|^2. \tag{9}$$

Assuming (4), and continuing from (9)

$$\frac{\alpha}{2C} \left| \log \frac{p(x_{1,N}|\theta')}{p(x_{1,N}|\theta)} - \log \frac{p(x_{2,N}|\theta')}{p(x_{2,N}|\theta)} \right|^2 \tag{10}$$

$$\leq \frac{\alpha}{2C} \left( \left| \log \frac{p(x_{1,N}|\theta')}{p(x_{1,N}|\theta)} \right| + \left| \log \frac{p(x_{2,N}|\theta')}{p(x_{2,N}|\theta)} \right| \right)^2 \tag{11}$$

$$\leq \frac{\alpha}{2C} |2B|^2 \tag{12}$$

$$\leq \frac{2\alpha B^2}{C}. \tag{13}$$

On the other hand, assuming (5), and again continuing from (9) gives

$$\frac{\alpha}{2C} \left| \log \frac{p(x_{1,N}|\theta')}{p(x_{1,N}|\theta)} - \log \frac{p(x_{2,N}|\theta')}{p(x_{2,N}|\theta)} \right|^2 \tag{14}$$

$$= \frac{\alpha}{2C} \left| \log \frac{p(x_{1,N}|\theta')}{p(x_{2,N}|\theta')} - \log \frac{p(x_{1,N}|\theta)}{p(x_{2,N}|\theta)} \right|^2 \tag{15}$$

$$\leq \frac{\alpha}{2C} \left( \left| \log \frac{p(x_{1,N}|\theta')}{p(x_{2,N}|\theta')} \right| + \left| \log \frac{p(x_{1,N}|\theta)}{p(x_{2,N}|\theta)} \right| \right)^2 \tag{16}$$

$$\leq \frac{\alpha}{2C} |2B|^2 \tag{17}$$

$$\leq \frac{2\alpha B^2}{C}, \tag{18}$$

which is the same bound as before.

$\square$

# 3 Proof of main text's Theorem 2

The Barker test amounts to checking the following condition:

$$\Delta^* + V_{nc} + V_{cor}^{(2)} > 0, \text{ where} \tag{19}$$

$$\Delta^* = \frac{N}{b} \sum_{i \in S} \underbrace{\log \frac{p(x_i|\theta')}{p(x_i|\theta)}}_{r_i} + \log \frac{q(\theta|\theta')p(\theta)}{q(\theta'|\theta)p(\theta')}, \tag{20}$$

$$V_{nc} \sim \mathcal{N}(0, 2 - s_{\Delta^*}^2), \tag{21}$$

$N$ is the full dataset size, $b$ is the batch size, $s_{\Delta^*}^2$ is the sample variance, and summation over $S$ here means summing over the elements in the batch, indexed by the element number $i$.

In other words, with a slight abuse of notation and writing capital letters for random variables the mechanism releases a sample from

$$\mathcal{N}(N\bar{\mathbf{r}}, 2 - \text{Var}(\frac{N}{b} \sum_{i \in S} R_i)) = \mathcal{N}(N\bar{\mathbf{r}}, 2 - \frac{N^2}{b^2} \sum_{i \in S} \text{Var}(R)) \tag{22}$$

$$\approx \mathcal{N}(N\bar{\mathbf{r}}, 2 - \frac{N^2}{b} \text{Var}(\mathbf{r})), \tag{23}$$

where (22) holds because $R_i$ are conditionally iid with a common distribution written as $R$, and $\text{Var}(\mathbf{r})$ means the sample variance estimated from the actual iid sample $r_i, i \in S$ we have, i.e., a vector of length $b$.

Assume that

$$|r_i| \leq \frac{\sqrt{b}}{N} \triangleq c, \forall i \text{ and} \tag{24}$$

$$\alpha < \frac{b}{5}. \tag{25}$$

We want to show that

$$D_\alpha(\mathcal{N}_1 \,\|\, \mathcal{N}_2) = \underbrace{\ln \frac{\sigma_2}{\sigma_1}}_{f_1} + \underbrace{\frac{1}{2(\alpha-1)} \ln \frac{\sigma_2^2}{\alpha\sigma_2^2 + (1-\alpha)\sigma_1^2}}_{f_2} + \underbrace{\frac{\alpha}{2} \frac{(\mu_1 - \mu_2)^2}{\alpha\sigma_2^2 + (1-\alpha)\sigma_1^2}}_{f_3} \tag{26}$$

$$\leq \frac{5}{2b} + \frac{1}{2(\alpha-1)} \ln \frac{2b}{b - 5\alpha} + \frac{2\alpha}{b - 5\alpha}. \tag{27}$$

*Proof.* As a first step, we have

$$0 < \mathrm{Var}(\mathbf{r}) = \mathbb{E}(\mathbf{r}^2) - \mathbb{E}(\mathbf{r})^2 \leq \mathbb{E}(\mathbf{r}^2) = 1/b \sum_{i \in S} r_i^2 \leq \frac{b}{N^2} \tag{28}$$

$$\Rightarrow 2 - \frac{N^2}{b} \mathrm{Var}(\mathbf{r}) \in [1, 2), \tag{29}$$

where the last inequality in (28) follows from (24).

Denote the maximally different adjacent datasets as $\mathbf{r}_1, \mathbf{r}_2$ that produce draws from $\mathcal{N}_1$ and $\mathcal{N}_2$ respectively, parameterised with means and variances as in (23). W.l.o.g., we can assume that the differing element is the final one, so we have $r_{1,i} = r_{2,i}, i = 1, \ldots, b-1$. We write $i \in S \setminus x_N$ to index a summation over the batch omitting the differing element.

The proof proceeds by bounding each of the terms $f_1, f_2, f_3$ in (26).

To start with, $f_1$ can be bounded as follows:

$$f_1 = \frac{1}{2} \ln \frac{\sigma_2^2}{\sigma_1^2} \leq \frac{1}{2} |\ln \frac{\sigma_2^2}{\sigma_1^2}| \leq \frac{1}{2} |\sigma_2^2 - \sigma_1^2| \tag{30}$$

$$= \frac{1}{2} |2 - \frac{N^2}{b} \mathrm{Var}(\mathbf{r}_2) - (2 - \frac{N^2}{b} \mathrm{Var}(\mathbf{r}_1))| \tag{31}$$

$$= \frac{N^2}{2b} |1/b \sum_{i \in S} r_{1,i}^2 - (\bar{\mathbf{r}}_1)^2 - 1/b \sum_{i \in S} r_{2,i}^2 + (\bar{\mathbf{r}}_2)^2| \tag{32}$$

$$= \frac{N^2}{2b} |1/b(r_{1,b}^2 - r_{2,b}^2) + (1/b \sum_{i \in S} r_{2,i})^2 - (1/b \sum_{i \in S} r_{1,i})^2| \tag{33}$$

$$= \frac{N^2}{2b^2} |(r_{1,b}^2 - r_{2,b}^2) + 1/b(r_{2,b}^2 - r_{1,b}^2 + 2(\sum_{i \in S \setminus x_N} r_{2,i} \cdot r_{2,b} - \sum_{i \in S \setminus x_N} r_{1,i} \cdot r_{1,b}))| \tag{34}$$

$$= \frac{N^2}{2b^2} |\frac{b-1}{b}(r_{1,b}^2 - r_{2,b}^2) - \frac{2}{b}(\sum_{i \in S \setminus x_N} r_i)(r_{1,b} - r_{2,b})| \tag{35}$$

$$= \frac{N^2}{2b^3} |(b-1)(r_{1,b}^2 - r_{2,b}^2) - 2(\sum_{i \in S \setminus x_N} r_i)(r_{1,b} - r_{2,b})| \tag{36}$$

$$\leq \frac{N^2}{2b^3} [(b-1)(c^2) + 2(b-1)c(2c)] \tag{37}$$

$$= \frac{N^2}{2b^3}(b-1)5c^2 \tag{38}$$

$$\leq \frac{5}{2b}, \tag{39}$$

where the final inequality in (30) holds because we have (29), and (37) as well as the final bound in (39) follow from (24).

For the common denominator term $\alpha\sigma_2^2 + (1-\alpha)\sigma_1^2$ in $f_2$ and $f_3$, we can first repeat essentially the previous calculation to get

$$\sigma_2^2 - \sigma_1^2 \geq -|\sigma_2^2 - \sigma_1^2| \tag{40}$$

$$= \cdots \tag{41}$$

$$= -\frac{N^2}{b^3}|(b-1)(r_{1,b}^2 - r_{2,b}^2) - 2(\sum_{i \in S\setminus x_N} r_i)(r_{1,b} - r_{2,b})| \tag{42}$$

$$\geq -\frac{N^2}{b^3}[(b-1)c^2 + 2(b-1)c(2c)] \tag{43}$$

$$= -\frac{N^2}{b^3}(b-1)5c^2 \tag{44}$$

$$\geq -\frac{5}{b}. \tag{45}$$

Combining (45) and (29) we get

$$\alpha\sigma_2^2 + (1-\alpha)\sigma_1^2 = \sigma_1^2 + \alpha(\sigma_2^2 - \sigma_1^2) \tag{46}$$

$$\geq 1 - \alpha\frac{5}{b} > 0, \tag{47}$$

where the final inequality follows from (25).

For the numerator in $f_3$ we have

$$(\mu_1 - \mu_2)^2 = \left(\frac{N}{b}\sum_{i \in S} r_{1,i} - \frac{N}{b}\sum_{i \in S} r_{2,i}\right)^2 \tag{48}$$

$$= \left(\frac{N}{b}(r_{1,b} - r_{2,b})\right)^2 \tag{49}$$

$$\leq \left(\frac{2Nc}{b}\right)^2 \tag{50}$$

$$\leq \frac{4}{b}. \tag{51}$$

Finally, using the derived bounds in (39), (47), and (51) with the fact that $\sigma_2^2 \leq 2$ from (29), the bound for the Rényi divergence (26) becomes

$$D_\alpha(\mathcal{N}_1 \| \mathcal{N}_2) \leq \frac{5}{2b} + \frac{1}{2(\alpha-1)}(\ln 2 - \ln(1 - \frac{5\alpha}{b})) + \frac{\alpha}{2}\frac{4}{b}\frac{1}{1 - \frac{5\alpha}{b}} \tag{52}$$

$$\leq \frac{5}{2b} + \frac{1}{2(\alpha-1)}\ln\frac{2b}{b - 5\alpha} + \frac{2\alpha}{b - 5\alpha}. \tag{53}$$

If we instead use the tempered log-likelihoods with temperature $\tau = \frac{N_0}{N}$, the effect is to replace $r_i$ by $\tau r_i$. The same proof then holds when instead of $N$ we write $N_0$.

$\square$

# 4    Bounding the approximations errors

As mentioned in the main text, with finite data and $b < N$ the acceptance test (18) in the main text is an approximation. For this case, there are some known theoretical bounds for the errors

induced. The general idea with the following Theorems is that by bounding the errors induced by each approximation step, we can find a bound on the error in the stationary distribution of the approximate chain w.r.t. the exact posterior. The references in this Section mostly point to the main text. The exceptions are obvious from the context.

First, Theorem 1 gives an upper bound for the error due to $\Delta^*$ having approximately normal instead of exactly normal distribution as in (20):

**Theorem 1.**
$$\sup_y |\mathbb{P}(\Delta^* < y) - \Phi(\frac{y - \Delta}{s_{\Delta^*}})| \leq \frac{6.4\mathbb{E}[|Z|^3] + 2\mathbb{E}[|Z|]}{\sqrt{b}},$$
*where* $Z = N(\log \frac{p(X|\theta')}{p(X|\theta)} - \mathbb{E}[\log \frac{p(X|\theta')}{p(X|\theta)}])$.

*Proof.* See [Seita et al., 2017, Cor. 1] . $\square$

Next, we have a bound for the error in the test quantity (18) relative to the exact test (5) given in Theorem 2. The original proof [Seita et al., 2017, Cor. 2] assumes that $C = 1$ and (16) holds exactly. We present a slightly modified proof that holds for any $C$ and also accounts for the error due to having only an approximate correction to the logistic distribution. We start with a helpful Lemma before the actual modified Theorem.

**Lemma 1.** *Let* $P(x)$ *and* $Q(x)$ *be two CDFs satisfying* $\sup_x |P(x) - Q(x)| \leq \epsilon$ *with* $x$ *in some real range. Let* $R(y)$ *be the density of another random variable* $Y$. *Let* $P'$ *be the convolution* $P * R$ *and* $Q'$ *be the convolution* $Q * R$. *Then* $P'(z)$ *(resp.* $Q'(z)$*) is the CDF of sum* $Z = X + Y$ *of independent random variables* $X$ *with CDF* $P(x)$ *(resp.* $Q(x)$*) and* $Y$ *with density* $R(y)$. *Then*
$$\sup_x |P'(x) - Q'(x)| \leq \epsilon.$$

*Proof.* See [Seita et al., 2017, Lemma 4] . $\square$

**Theorem 2.** *If* $\sup_y |\mathbb{P}(\Delta^* < y) - \Phi(\frac{y-\Delta}{s_{\Delta^*}})| \leq \epsilon_1(\theta', \theta, b)$ *and* $\sup_y |S'(y) - S(y)| \leq \epsilon_2$, *then* $\sup_y |\mathbb{P}(\Delta^* + V_{nc} + V_{cor}^{(C)} < y) - S(y - \Delta)| \leq \epsilon_1(\theta', \theta, b) + \epsilon_2$, *where* $s_{\Delta^*}$ *is the sample standard deviation of* $\Delta^*$, $S'$ *is the cdf of the approximate logistic distribution produced by* $\mathcal{N}(0, C) + V_{cor}^{(C)}$, *and* $S$ *is the exact logistic function.*

*Proof.* As in the original proof [Seita et al., 2017, Cor. 2] the main idea is to use Lemma 1 two times. First, take $P(y) = \mathbb{P}(\Delta^* < y), Q(y) = \Phi(\frac{y-\Delta}{s_{\Delta^*}})$ and convolve with $V_{nc}$ which has density $\phi(\frac{x}{\sqrt{C-s_{\Delta^*}^2}})$. For the second step, take the results $P'(y) = \mathbb{P}(\Delta^* + V_{nc} < y), Q'(y) = \Phi(\frac{y-\Delta}{\sqrt{C}})$ and convolve with the density of $V_{cor}^{(C)}$ to get $P''(y) = \mathbb{P}(\Delta^* + V_{nc} + V_{cor}^{(C)} < y), Q''(y) = S'(y - \Delta)$. By Lemma 1, both convolutions preserve the error bound $\epsilon_1(\theta', \theta, b)$, and we therefore have

$$\sup_y |\mathbb{P}(\Delta^* + V_{nc} + V_{cor}^{(C)} < y) - S(y - \Delta)| \tag{54}$$

$$= \sup_y |\mathbb{P}(\Delta^* + V_{nc} + V_{cor}^{(C)} < y) - S'(y - \Delta) + S'(y - \Delta) - S(y - \Delta)| \tag{55}$$

$$\leq \sup_y |\mathbb{P}(\Delta^* + V_{nc} + V_{cor}^{(C)} < y) - S'(y - \Delta)| + \sup_y |S'(y) - S(y)| \tag{56}$$

$$\leq \epsilon_1(\theta', \theta, b) + \epsilon_2, \tag{57}$$

where (56) follows from the triangle inequality. $\square$

Finally, a bound on the test error implies a bound for the stationary distribution of the Markov chain relative to the true posterior, given in Theorem 3. Writing $d_v(P, Q)$ for the total variation distance between distributions $P$ and $Q$, $\mathcal{T}_0$ for the transition kernel of the exact Markov chain, $\mathcal{S}_0$ for the exact posterior, and $\mathcal{S}_\epsilon$ for the stationary distribution of the approximate transition kernel where $\epsilon$ is the error in the acceptance test, we have:

**Theorem 3.** *If $\mathcal{T}_0$ satisfies the contraction condition $d_v(P\mathcal{T}_0, \mathcal{S}_0) < \eta d_v(P, \mathcal{S}_0)$ for some constant $\eta \in [0, 1)$ and all probability distributions $P$, then*

$$d_v(\mathcal{S}_0, \mathcal{S}_\epsilon) \leq \frac{\epsilon}{1 - \eta},$$

*where $\epsilon$ is the bound on the error in the acceptance test.*

*Proof.* See [Korattikara et al., 2014, Theorem 1] . $\qquad\qquad\square$

Generally, especially the contraction condition in Theorem 3 can be hard to meet: it can be shown to hold e.g. for some Gibbs samplers (see e.g. Brémaud 1999, Theorem 6.1) but it is not usually valid for an arbitrary model, and even checking the condition might not be trivial.

# 5    Numerical approximation of the correction distribution

As noted in the main text, we need to find an approximate distribution $V_{cor}^{(C)}$ s.t.

$$V_{log} \overset{d}{=} \mathcal{N}(0, C) + V_{cor}^{(C)}, \tag{58}$$

where $V_{log}$ has a standard logistic distribution. The approximation method of Seita et al. [2017] casts the problem into a ridge regression problem, which can be solved effectively. However, nothing constrains the resulting function from having negative values. In order to use it as an approximate pdf, Seita et al. [2017] set these to zeroes and note that as long as $C$ is small enough, such values are rare and hence do not affect the solution much. In practice, their solution seems to work very well with small values of $C$, e.g. when $C \leq 1$.

Since we want to use larger $C$ for the privacy, we propose to approximate $V_{cor}^{(C)}$ with a Gaussian mixture model (GMM). Since the result is always a valid pdf, the problem of negative values does not arise.

To find the correction pdf, denote the density of the GMM approximation with $K$ components by $\tilde{f}_{cor}$, the GMM component parameters by $\pi_k$, $\mu_k$ and $\sigma_k$, and the standard normal density by $\phi$. We have

$$
\begin{aligned}
f_{log}(x) &= (f_{norm} * f_{cor})(x) \simeq (f_{norm} * \tilde{f}_{cor})(x) \\
&= \int_{\mathbb{R}} f_{norm}(x)\tilde{f}_{cor}(x - t)dt \\
&= \int_{\mathbb{R}} \phi(\frac{x}{\sqrt{C}})[\sum_{k=1}^{K} \pi_k \phi(\frac{x - t - \mu_k}{\sigma_k})]dt \\
&= \sum_{k=1}^{K} \pi_k \phi(\frac{x - \mu_k}{\sqrt{C + \sigma_k^2}}) = \tilde{f}_{log}^{(C)}(x; \pi_k, \mu_k, \sigma_k, k = 1, \ldots, K)
\end{aligned}
$$

As the logistic pdf is symmetric around zero, we require our GMM approximation to be symmetric as well. We achieve this by creating a counterpart for each mixture component with an opposite sign mean and identical variance and weight. To construct the approximation on some interval $[-a, a] \subset \mathbb{R}$, we discretise the interval into $n$ points, and fit the GMM by minimising the loss function

$$\mathcal{L}(\pi, \mu, \sigma) = \|f_{log} - \tilde{f}_{log}^{(C)}\|_2 \qquad (59)$$

calculated over the discretisation. Since GMM is a generative model, sampling from the optimised approximation is easy.

Figure 1 shows the approximation error $\max_y |S'(y) - S(y)|$, where $S'$ is the approximate logistic ecdf and $S$ the exact logistic cdf, due to $\tilde{V}_{cor}$ using the ridge regression solution proposed by Seita et al. [2017] and the GMM. The error measure is the same as in Theorem 2 in the Supplement. Empirically, as shown in the Figure, we can have noticeably better approximation especially with larger $C$ values.

**Figure 1:** *Approximation error due to $\tilde{V}_{cor}$ with error bars showing the standard error of the mean calculated from 20 runs. With the ridge regression solution proposed by Seita et al. [2017] the error increases quickly when $C > 1$. Using the GMM approximation we can achieve significantly smaller error with $C = 2$.*

Figure 2 shows the two approximations with increasing $C$. When the negative values in the ridge regression solution are projected to zeroes, the variance of $V_{cor}$ increases and the resulting approximate $\tilde{V}_{log}$ has variance much larger than the actual $\pi^2/3$ it should have. This also shows in the resulting approximation. Figure 3 shows the empirical cdf for both approximations and for the true logistic distribution, and the absolute distance between the approximations $S'$ and the true logistic cdf $S$.

To calculate the ridge regression solution for $[-10, 10]$, we use the original code of Seita et al. [2017] with parameter values $n = 4000, \lambda = 10.0$ used in the original paper. The problems with larger $C$ values persisted with other parameter settings we tested. Note that the discretisation granularity parameter $n$ used in the two methods are not directly comparable.

To fit the GMMs with $K$ components, we take the interval $[-10, 10]$ with $n = 1000$ points for calculating the loss function, and run 20000 optimisation iterations with PyTorch [Paszke et al., 2017]. We use Adam optimiser [Kingma and Ba, 2014] with learning rate $\eta = 0.01$ and otherwise default settings. The approximation is forced to be symmetric about zero by adding mirrored components: for the $k$th component we add a copy but set the mean as $-\mu_k$, and set

Figure 2: *Approximate correction distribution log-densities with varying $C$ values. Figure 2(a) shows the results for the ridge regression solution used by Seita et al.: as $C$ increases, the amount of negative values that are projected to zeroes, which show as gaps in the log-pdf, increases markedly. Figure 2(b) shows corresponding results for our GMM solution: the approximation is always a valid pdf over $\mathbb{R}$.*

the weights as $\pi_k/2$ for both, i.e., use the mean of the original and the mirrored component. We use $K = 50$ in the test, which gives 100 components with mirroring.

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

Yu-Xiang Wang, Borja Balle, and Shiva Prasad Kasiviswanathan. Subsampled Rényi differential privacy and analytical moments accountant. In Kamalika Chaudhuri and Masashi

Sample ecdf for $C = 2.0$ and true logistic cdf
$X_{cor} + \mathcal{N}(0, C)$ estimated with 50000 samples

Abs differences between approximations and true logistic cdf for $C = 2.0$
$X_{cor} + \mathcal{N}(0, C)$ estimated with 50000 samples

(a) Approximation ecdf and true logistic cdf

(b) Absolute differences from true logistic cdf

**Figure 3:** *Figure 3(a) shows the empirical cdf for the approximate logistic distributions calculated using the ridge regression solution of Seita et al. and our GMM together with true logistic cdf. The variance of $V_{cor}$ using ridge regression is too high and the resulting $V_{cor} + \mathcal{N}(0, C)$ is clearly off. The ecdf for GMM is almost indistinguishable from the true cdf. Figure 3(b) shows the absolute distances between the approximation ecdf and the true logistic cdf.*

Sugiyama, editors, *Proceedings of Machine Learning Research*, volume 89 of *Proceedings of Machine Learning Research*, pages 1226–1235. PMLR, 16–18 Apr 2019.