[Reviews · NeurIPS 2019]

Reviewer 1



I do not have many comments, as the paper is technically strong, and the writing is extremely clear. The authors connect very well to previous work. The experimental results are useful to have as well. Thank you for your rebuttal. Overall, even though the paper is well-written, the use of the CLT makes some of the proofs a bit unsatisfactory and hard to check: checking requires a bit more intuition than I would want in a formal context. In the end the paper is making a number of approximations, but since they appear to be useful in practice, I think it is fine as long as the authors emphasize where approximations are made and how their theoretical results could be strengthened,

Reviewer 2



This work provides a detailed Renyi DP analysis of a modified MCMC acceptance test, and empirically demonstrates its efficacy. Originality: the RDP analysis and modified acceptance test is a novel contribution. Quality: the work is a complete piece on exploring this MCMC method, with a detailed analysis and experiments. Clarity: the work is fairly clearly written, but it can be easy to lose track of exactly what parameters remain as choices to be tuned in a list of various corrective factors and approximations. Significance: the work gives an MCMC method with privacy without convergence, which permits privacy guarantees to be given over a multitude of problems without doubts or guess work about when to stop the chain.

Reviewer 3



The paper is proposing a DP MCMC algorithm for posterior sampling, adopting the Barker acceptance decomposition and sub-sampling, and by that improving the DP guarantees and the computational requirements. The paper has a rigorous language and it is nicely written. What needs to be more clear from the beginning is the contribution over Seita et al. It takes to page 4 to figure out what this paper is contributing, which is all the results on differential privacy under decomposition and sub-sampling. In addition, the paper needs to clarify what is the aim of having privacy and against which third party?

[Author Response · NeurIPS 2019]

We thank the reviewers for their positive feedback and will revise the paper accordingly. The following are answers to the questions posed by each reviewer:

**Reviewer 1**

The CLT assumption does not affect the privacy of the method. If the assumption is heavily violated during an MCMC run, the method still guarantees privacy but the result might be far from the true posterior. We do not treat the CLT in more depth in the paper since this is one of the central themes of Seita et al. 2017, and we do not claim any contribution to this theory. This is clearly one of the most interesting questions for further research.

**Reviewer 2**

We will include a list of the different factors to improve clarity:

1. Common parameters
    (a) $\alpha$ parameter for RDP.
    (b) $T$ number of iterations.
    (c) $N$ full data size.
2. DP MCMC with full data
    (a) $C \in (0, \pi^2/3)$ freely chosen constant for noise variance: higher $C$ improves privacy but also increases decomposition error.
    (b) $B$ assumed bound for the log-likelihood ratios (llr) w.r.t. data OR the parameters. This holds for Lipschitz llr, or it can be enforced by using small enough proposal variance (for suitable models), or by clipping (for any arbitrary model). Smaller $B$ improves privacy, while smaller proposal variance means the chain moves slower, and tighter clipping means less accurate posterior estimation.
3. DP MCMC with subsampling
    (a) $b$ batch size s.t. $\alpha < b/5$: smaller $b$ means better privacy amplification but generally also more error in the CLT approximation.
    (b) Set noise variance $C = 2$ (analysis could be done for other values as well).
    (c) Assume llr w.r.t. the parameters $\leq \sqrt{b}/N$
        OR
        optionally with tempering: choose $\beta$ in the coarsened posterior s.t. $N_0 = N\beta/(\beta + N)$ and assume llr w.r.t. the parameters $\leq \sqrt{b}/N_0$.
        As above, either condition holds for Lipschitz llr or can be enforced by tuning the proposal variance or by clipping.

We agree that a good comparison of different approaches to DP samplers would be interesting. However, we feel that this is out of scope for the current submission, since the various methods have different assumptions.

**Reviewer 3**

We will clarify our contribution as opposed to the Seita et al. 2017 paper more clearly starting already from the Introduction.

The decomposition idea is discussed more extensively by Seita et al. 2017 (for achieving subsampling without any privacy notion). In our paper, most of the discussion regarding the decomposition is in the Supplement, while in the main paper we focus more on the privacy.

The privacy setting we consider is the standard centralised setting for DP: a single trusted party (data curator) has access to all the data and runs the MCMC algorithm with the aim of releasing a trained DP model and/or the samples from the chain while protecting the privacy of the training data (with one sample corresponding to one individual). The adversary has access to the trained model/samples from the chain and (almost) arbitrary side information as per DP definition. We will clarify this in the final version.

[Meta-Review · NeurIPS 2019]

This paper advances the state-of-the-art in differentially private Bayesian ML by proposing a private MCMC sampling scheme with wide applicability. Although the analysis follows some known ideas from the literature on private SGD, there are a number of new tricks which make the current approach interesting, most notably the observation that one can use randomized acceptance tests to preserve privacy in an MCMC algorithm. When preparing the final version of this manuscript the authors should carefully consider the points raised in the reviews regarding: clarifying where the contributions lie with respect to previous work; provide high-level intuitions of the proofs to help a reader navigate the derivations; discuss the role of approximations used in the paper, where they affect the privacy or utility of the method, and where there is some room left for improvement.